# Synthesis, Photophysics, and Solvatochromic Studies of an Aggregated-Induced-Emission Luminogen Useful in Bioimaging

**DOI:** 10.3390/s19224932

**Published:** 2019-11-12

**Authors:** Laura Espinar-Barranco, Marta Meazza, Azahara Linares-Perez, Ramon Rios, Jose Manuel Paredes, Luis Crovetto

**Affiliations:** 1Department of Physical Chemistry, University of Granada, Unidad de Excelencia en Quimica Aplicada a Biomedicina y Medioambiente (UEQ), Cartuja Campus, 18071 Granada, Spain; lauraebe@correo.ugr.es; 2School of Chemistry, University of Southampton, Highfield Campus, Southampton SO17 1BJ, UK; M.Meazza@soton.ac.uk (M.M.); R.Rios-Torres@southampton.ac.uk (R.R.); 3Department of Biochemistry and Molecular Biology II, Faculty of Pharmacy, University of Granada, Cartuja Campus, 18071 Granada, Spain; azaharalinares@ugr.es

**Keywords:** aggregated enhanced emission, photophysics, bioimaging

## Abstract

Biological samples are a complex and heterogeneous matrix where different macromolecules with different physicochemical parameters cohabit in reduced spaces. The introduction of fluorophores into these samples, such as in the interior of cells, can produce changes in the fluorescence emission properties of these dyes, caused by the specific physicochemical properties of cells. This effect can be especially intense with solvatofluorochromic dyes, where changes in the polarity environment surrounding the dye can drastically change the fluorescence emission. In this article, we studied the photophysical behavior of a new dye and confirmed the aggregation-induced emission (AIE) phenomenon with different approaches, such as by using different solvent proportions, increasing the viscosity, forming micelles, and adding bovine serum albumin (BSA), through analysis of the absorption and steady-state and time-resolved fluorescence. Our results show the preferences of the dye for nonpolar media, exhibiting AIE under specific conditions through immobilization. Additionally, this approach offers the possibility of easily determining the critical micelle concentration (CMC). Finally, we studied the rate of spontaneous incorporation of the dye into cells by fluorescence lifetime imaging and observed the intracellular pattern produced by the AIE. Interestingly, different intracellular compartments present strong differences in fluorescence intensity and fluorescence lifetime. We used this difference to isolate different intracellular regions to selectively study these regions. Interestingly, the fluorescence lifetime shows a strong difference in different intracellular compartments, facilitating selective isolation for a detailed study of specific organelles.

## 1. Introduction

When one dye has weak emission in dilute solutions but its aggregation causes strong emission, we call this behavior aggregation-induced emission (AIE). Moreover, the aggregation effects can include changes on other dye fluorescence properties such as shifts in its spectral position. For example, this effect has been observed in a coumarin-based fluorogen [1] and acridine orange in reverse micelles [2] or porphyrins [3]. Since its discovery in 2001 [4], this phenomenon has received much attention [5,6,7] and has been developed in several applications, such as organic light-emitting devices [8], bioimaging [9], ion sensors [10,11], and environmental analysis [12]. AIE is therefore a very interesting behavior present in many molecules, with multiple uses in different scientific fields.

Biological samples are a complex and heterogeneous matrix where different macromolecules with different physicochemical parameters cohabit in reduced spaces. The introduction of a fluorophore in these samples, such as in the interior of cells, can produce changes in the fluorescent emission properties of these dyes caused by the specific physicochemical properties of cells [13]. This effect can be especially intense with solvatofluorochromic dyes, where the changes in the polarity environment surrounding the dye can drastically change the fluorescence emission [14]. The crowded environment typical of intracellular media can produce AIE by restricting the intramolecular motion of the dye due to immobilization upon interaction with intracellular macromolecules. AIE makes these molecules excellent probes in the fields of biomedicine and life sciences [15]. AIE probes have been used for biosensing and/or bioimaging in DNA [16] and ATP [17] detection, as a mitochondria marker [18], for determining lipid droplet and mitochondria membrane potential [19], and for detecting multiple ions and species such as Al^3+^ in food and cells [20], intracellular cyanide [21], bilirubin and Fe^3+^ [22]. These molecules present a promising future as probes in in vivo diagnosis and therapy [23].

The study and development of new AIE luminogens is currently an interesting research topic with important potential uses. However, prior to application, it is important to understand their photophysical and solvatochromic behavior. This consideration is especially important in fluorescence lifetime imaging microscopy (FLIM) for correct interpretation of the data obtained. FLIM is an innovative multiparametric tool in which intensity and fluorescence lifetime are recovered in an image. Apart from intensity, the fluorescence lifetime can reveal different environments and be affected by analytes [24,25,26] but with the advantage of being independent of the luminogen concentration. In this article, we have synthesized and studied the solvatochromic properties of the dye 2-((4-((E)-2-((E)-3-((E)-3-(4-cyano-5-(dicyanomethylene)-2-phenyl-2-(trifluoromethyl)-2,5-dihydrofuran-3-yl)allylidene)-5,5-dimethylcyclohex-1-en-1-yl)vinyl)phenyl)(ethyl)amino)ethyl methacrylate (PEMC) (see Scheme 1 and Appendix A) using different approaches such as the Catalan and Lippert–Mataga analyses. We also measured the effect on fluorescence-forming sodium dodecyl sulfate (SDS) micelles and added bovine serum albumin (BSA), analyzing the absorption and steady-state and time-resolved fluorescence. Finally, we measured the behavior in live cells through FLIM.

## 2. Results

### 2.1. Solvatochromism

An initial solvatochromic study using 14 different solvents was performed (Figure 1). We measured the absorption, emission, and fluorescence lifetime of the dye PEMC. We observed a strong dependency on the nature of the solvent used, with maxima shifts of 57 and 50 nm of dye PEMC absorption and emission, respectively.

Moreover, we have calculated the relative quantum yield in every solvent using Nile Blue A as a reference. The main spectral parameters and quantum yield are included in Table 1.

As solvatochromism is essential to understand the behavior of dyes in biological samples and allows us to determine a correct interpretation of the fluorescence signal, we performed a study using the Catalan approach [29] and the Lipper–Mataga equation. The Lipper–Mataga equation reveals a slight decrease in the spectral shift and polarizability (Δf; see Appendix A), a term that represents the changes in the dipole moment and energy after excitation. This term includes the dielectric constant and refractive index of the solvent. However, the deeper and stronger Catalan approach can reveal the influence of four physicochemical parameters (acidity, basicity, dipolarity, and polarizability) on the changes in absorption and emission spectral properties. Our study revealed a good relationship between the experimental and theoretical data using the four parameters obtained from the Catalan approach (see Appendix A). After removing the parameters one by one, the parameter responsible for decreasing the good relationship is determined to be the critical physicochemical parameter that affects the spectral shift in absorption and emission. Appendix A reveals the fitting of the Catalan approach. As can be observed, for absorption and emission, we obtain *r* values supporting good correlations (0.9420 and 0.9736, respectively) that remain high when we remove every parameter except dipolarity, which reduces the values to 0.6602 and 0.6175. To confirm that dipolarity is the only parameter that affects solvatochromism, we also obtained data using only one physicochemical parameter. As can be observed in Appendix A, our fitting shows low *r* values, except when we used the dipolarity parameter, which remains high, indicating a good correlation (0.8856 and 0.9538 for absorption and emission, respectively). Therefore, our results show that dipolarity is the main physicochemical parameter that affects the spectral shift in absorption and emission. This finding is in good accordance with the Lipper–Mataga representation because dipolarity is not included in the polarizability term (Δf). For this reason, the Lipper–Mataga representation does not show any strong change with polarizability.

As expected, for solvathocromic dyes, the dependence of the fluorescence properties on the dipolarity can be used as sensing in bioimaging, because inside cells, it can be found important changes in the dipolarity in the microenvironments. The increase in fluorescence from water to nonpolar solvents makes this molecule an interesting probe to detect its incorporation into nonpolar structures (micelles, proteins, membranes, etc.) from aqueous solvent. Moreover, these structures can cause a restriction in the intramolecular motion of the dye, leading to AIE.

### 2.2. Aggregation-Induced Emission

We checked the enhancement of fluorescence intensity by aggregation increasing the concentration of PEMC in ethanol. Figure 2a shows a plot of maxima intensities at different concentrations and spectra are in Appendix A. At lowest concentrations we observed a linear increase in intensity (as predicted by Kavanagh’s law). This linearity is broken when concentration becomes high enough. However, at very high concentrations, aggregation happens, enhancing strongly the fluorescence intensity. In the other hand, to evaluate whether dipolarity is also key in the AIE process, we measured PEMC in different polarity environments containing different proportions of water: THF. As can be observed in Figure 2b and Appendix A, the intensity rapidly increases in the presence of a low percentage of THF, with a maximum at 2.5% THF, decreases from 5% to 30% THF, and then increases slightly until 100% THF. Because in THF, the quantum yield of PEMC is higher than that in water, the fluorescence measured is higher in THF than in water. The fast increase at 2.5% THF can be explained as follows. The molecules start to form small aggregates as a result of electrostatic interaction with the solvent mixture leading to AIE. When the proportion of THF molecules increases, aggregates are broken or form even larger aggregates which are non-fluorescent probably as a result of aggregation induced quenching. PEMC becomes more widely distributed and therefore loses its local aggregation, causing decreased fluorescence due to AIE. So, 2.5% is the critical concentration when AIE is maximum.

Another approach to corroborate the AIE behavior is by limiting the rotational freedom of the dye, such as by increasing the viscosity of the solvent or studying incorporation of the dye into a macromolecular structure. In the first case, we added glycerin to increase the viscosity of the sample. Figure 3a shows the increase in fluorescence when 25% and 50% glycerin is present in the solution, indicating that PEMC can also be considered as a probe of viscosity.

In contrast, AIE can be used to observe the spontaneous incorporation of macromolecules such as proteins. To follow this behavior, we added BSA to a previously prepared solution of PEMC. Two additions of BSA were performed. As observed in Figure 3b, the incorporation of 20 μg of BSA produces a two-fold fluorescence intensity increase, and the addition of 50 μg of BSA produces a stronger effect, enhancing the fluorescence intensity by ten-fold.

Based on the previous data and according to the characteristic that the dye fluorescence is highly dependent on the surrounding dipolarity and exhibits AIE when immobilized, we hypothesized to use the dye as a probe to detect micelle formation. For this purpose, SDS was selected as it can be mixed at different concentrations to study the influence on PEMC before and after micelle formation. Figure 4a shows the fluorescence intensity of PEMC at different SDS concentrations. The dye is almost nonfluorescent when it is dissolved in water or even at an SDS concentration lower than 1 mM. However, above an SDS concentration of 3 mM, and until 5 mM, the fluorescence of PEMC is enhanced. This increase in the fluorescence is attributed to the incorporation of the dye molecules into the SDS micelles when they are formed. Therefore, the solvent can be used to determine the CMC and the formation of micelles when it is confined in this small volume to produce AIE. At higher CMC values, the fluorescence intensity exhibited a slight decrease, as shown in Figure 4b.

All these characteristics of PEMC make it useful in further applications for bioimaging. As the interiors of cells are characterized by a complex matrix, AIE can be considered an advantageous behavior to identify and as a marker of internal organelles. For this reason, we performed FLIM, studying the rate of spontaneous introduction and the internal fluorescence pattern of PEMC.

### 2.3. FLIM

First, we verified whether PEMC spontaneously enters the interior of cells by measuring the incorporation into the MDA-MB-231 cell line under experimental conditions where autofluorescence is negligible. Figure 5a shows the kinetics of intracellular fluorescence after the addition of 5 µL of PEMC (EtOH) in 1 mL of phosphate buffered saline (PBS) for a final concentration of 5 × 10^−6^ M PEMC. As can be observed, the incorporation is spontaneous, but it takes a period of time to achieve equilibrium, approximately 70 min (see Figure 5b). FLIM can recover the fluorescence lifetime and when a sufficient level of intensity is achieved to collect this signal, this parameter is independent of the dye concentration. We have observed that after 10 min, we obtain an interesting intracellular pattern where different structures are well defined in the intensity images (see Figure 5a), and marked with higher fluorescence lifetimes in FLIM images (see Appendix A). Appendix A collects the average fluorescence lifetime from Figure 5.

In addition to the MDA-MB-231 cell line, we used HEK cells to observe the intracellular pattern as much in intensity as in the fluorescence lifetime. Interestingly, peripheral F-actin structures at the plasma membrane in HEK293 cells [30] present a low intensity signal but a higher fluorescence lifetime, as shown in Figure 6a and Appendix A. Peripheral F-actin is clearly identifiable in this cell line, because their typical structures around the cells avoid any mistake with other organelles [30]. Respecting other intracellular structures, an enhancement of the fluorescence intensity is observed, with a corresponding higher value of the fluorescence lifetime. A similar intracellular pattern (with the exception of the peripheral F-actin structures) is found in the MDA-MB-231 cells (Figure 6b).

AIE is intracellularly produced in some internal structures. The difference in intensity can be used to isolate different organelles based on the intensity threshold. Briefly, as different structures have different intensities values, we used intensity images to select different regions of interest (ROI) through appropriate thresholds (see Appendix A). The ROI selected can be used to isolate these regions in the fluorescence lifetime images and obtain the fluorescence lifetime of PEMC in these structures. Therefore, this approach can efficiently study independent organelles based on fluorescence lifetime. Figure 7a shows HEK cells segmented with the resulting fluorescence lifetime image maps and histograms. Interestingly, we observed that three different isolated regions of the cells present well-differentiated fluorescence lifetimes (Figure 7b).

To go further in the investigation of these intracellular structures, we used a commercial mitochondria biomarker. Following an alternate excitation (470 nm to mitochondria biomarker and 635 nm to PEMC), and recovering their fluorescence in two different channels (green for mitochondria biomarker and red for PEMC), we selected the pixel where in both channels collocated to isolate the mitochondria and studied the fluorescence lifetime in these compartments. Appendix A shows the co-localization images, the complete FLIM image, and the mitochondria-isolated FLIM images. As can be observed, the mitochondria biomarker presents some artifacts marking some structures that are not mithocondria (placed in peripheria). These artifacts are easily identifiable by studying the mitochondria-isolated FLIM images because they show a different fluorescence lifetime.

## 3. Discussion

AIE is an interesting phenomenon that can be used as a tool to determine different biological and/or biomedical mechanisms, including intracellular processes. However, the use of AIE requires the complete knowledge of the photophysical properties of a given dye for proper interpretation and use. For this reason, we first realized the photophysical characterization of the dye diluted in different solvents following two different approaches, the Lipper–Mataga equation and the Catalan approach. From our analysis, the Lipper–Mataga analysis does not show any dramatic changes, except for a slight decrease in polarizability (Δf), a parameter that takes into account the effect of the solvent to change the dipole moment of the dye. However, the deeper Catalan approach allows the discrimination of which of four different physicochemical parameters (acidity, basicity, polarizability and dipolarity) affects the absorption and emission behavior. Following this analysis, we found that dipolarity is the only parameter that affects the behavior. Dipolarity is the capacity of the solvent to contain a dipolar system. Based on this result, and according to the data from the photophysical characterization in which the dye presents different relative quantum yields in the solvents, to verify the AIE, we mixed different proportions of water and THF. At a very low THF proportion (1%), we found a strong increase in the fluorescence signal. We explain this behavior as follows: because PEMC presents a higher affinity for this solvent, it accumulates around THF molecules, and therefore, the dye molecules become very closely packed, producing AIE. However, increasing the THF proportion produces a decrease in the intensity because the higher amount of THF molecules allows PEMC to disperse widely, losing its aggregation state and, therefore, the AIE behavior. When the THF proportion is further increased, the fluorescence intensity slightly increases due to the higher quantum yield of the dye in this solvent. To confirm aggregation is responsible of the enhancement of fluorescence, we increased the concentration of PEMC using ethanol as a solvent until it achieved aggregation. Once this condition was obtained, we observed a very strong enhancement of fluorescence.

Another approach to confirm the AIE is the immobilization of the dye by increasing the viscosity or binding it to certain substances. This is a very interesting behavior with useful applications in biological and biomedical fields. To observe the effect of the immobilization of the dye, we realized both approaches, first by increasing the solvent viscosity, and then, by taking advantage of the spontaneous incorporation of PEMC into nonpolar media, we used SDS molecules to study the behavior of the dye before and after the formation of micelles. Increasing the viscosity has a strong effect on fluorescence intensity. The addition of glycerol to the solvent produces a strong enhancement of the intensity. In the other approach, adding SDS to the solvent, below the CMC, the dye exhibits increased fluorescence due to the higher proportion of SDS in the medium. This behavior is probably due to the same effect previously discussed regarding the addition of THF. Once the CMC is reached and micelles are spontaneously formed, the dye exhibits its maximum fluorescence intensity due to AIE caused by the immobilization of the dye. After further increasing the SDS proportion, a slight decrease in the fluorescence intensity of PEMC was observed. This result occurs because a higher proportion of SDS can produce micelles with higher volumes in which the dye can become partially less immobilized. To confirm that this approach can be extended to study changes in the conformation or denaturation of proteins, we also added BSA protein to a solution of PEMC. We performed two different additions: 30 and 50 μg of BSA. We observed that when BSA is incorporated, the fluorescence of the dye increases drastically due to the self-introduction of the dye within the less polar interior of the protein, causing immobilization of the dye or even a self-interaction of the dye molecules in the interior of the proteins. Both cases cause AIE, with a ten-fold increase in the fluorescence intensity when 50 μg of BSA is added to the solution.

Finally, we studied the time of spontaneous incorporation of PEMC into live cells. In contrast to xanthenic dyes [14,25,26,31] (where the incorporation is very fast), PEMC achieves equilibrium approximately 70 min after addition of this luminogen. After this period of time, our data showed a plateau in the intracellular intensity signal. Following the fluorescence lifetime images, the lifetime remains almost invariable due to the concentration independence of this parameter. However, it takes approximately 20 min to observe a characteristic pattern where intracellular structures are clearly marked with higher fluorescence lifetime. This effect is probably due to AIE, where the PEMC molecules are immobilized in these structures, achieving a higher fluorescence intensity and a higher fluorescence lifetime. More interesting is the potential application in image analysis that can be performed owing to AIE behavior inside the cells. The differences in the intensity fluorescence can be used to isolate intracellular compartments in different regions of the cell. For this purpose, we isolated three different regions from HEK cells (see Figure 7). First, we selected the peripheral F-actin, where the luminogen presents lower intensity than that in other regions but a very high fluorescence lifetime. In the second and third regions, we differentiated between two different intracellular compartments. In the second region, PEMC exhibits very high intensity and intermediate fluorescence lifetime, and the third segmented region is characterized by multiple organelles with intermediate intensity but lower fluorescence lifetimes. For better characterization of the sensing aspect of this dye, we used a mitochondria biomarker to specifically isolate these compartments and measured the fluorescence lifetime in these organelles (Appendix A).

## 4. Materials and Methods

The dye PEMC was dissolved in EtOH, and 15 µL was added to a final volume of 1500 µL to different solvents with a final concentration of 5 × 10^−6^ M. The solvents used were butanol, acetone, cyclohexane, 1,4-dioxane, ethanol, methanol, dichloromethane, tetrahydrofuran, dibutyl ether, diethyl ether, chlorobenzene, toluene, propanol, dimethyl sulfoxide, acetonitrile, and acidic and basic water.

### 4.1. Instrumentation

Absorption spectra were performed on a Lambda 650 UV-visible spectrophotometer (PerkinElmer, Waltham, MA, USA). Fluorescence emission spectra were collected on a Jasco FP-8300 spectrofluorometer (Jasco, Tokyo, Japan) at the maximum absorption wavelength for each solvent. Fluorescence quantum yields were obtained using Nile Blue A as a reference (see Appendix A). Fluorescence lifetime decays were recorded by the single-photon timing method using a FluoTime 200 fluorometer (PicoQuant GmbH, Berlin, Germany). The excitation source was a pulsed diode laser at λ = 635 nm LDH-635, (PicoQuant), at a repetition rate of 40 MHz. Fluorescence decay histograms were collected at the maximum emission of each solvent, between 640 and 680 nm over 1320 channels, with increments per channel of 37 ps. Histograms of the instrument response functions (using a LUDOX scatterer) and sample decays were recorded until they reached 2 × 10^4^ counts in the peak channel. The fluorescence decay traces were globally analyzed using an iterative deconvolution method with exponential models using FluoFit software (PicoQuant).

### 4.2. Cell Culture

Embryonic kidney cells HEK-293 (ATCC no. CRL-1573™) were grown at 37 °C in Dulbecco’s modified Eagle’s medium (DMEM) supplemented with 10% (*v*/*v*) fetal bovine serum (FBS), 2 mM glutamine and 100 U/mL penicillin. Cell cultures were maintained in an incubator at 37 °C with 95% humidity and 5% CO2.

For the FLIM experiments, cells were seeded onto circular coverslips (diameter of 25 mm) in six-well plates at a density of 2.3 × 10^5^ cells per well.

### 4.3. FLIM Bioimaging

Images of fluorescence emission intensities and fluorescence lifetime were performed on a MicroTime 200 fluorescence-lifetime microscope system (PicoQuant GmbH, Berlin, Germany) using the same excitation source described earlier. The light beam was directed through a dichroic mirror (PIE 470/635 dcxr, Chroma) and through an oil immersion objective (1.4 NA, 100×) specific to an inverted microscope system ((IX-71, Olympus, Tokyo, Japan). The fluorescence emission was carried to a 500-nm long-pass filter (AHF/Chroma) and focused to a 75-µm confocal aperture. The fluorescence emission passed through a bandpass filter (D685/70M, Chroma) and focused on single-photon avalanche diodes (SPCM-AQR 14, Perkin Elmer). The data were obtained by a TimeHarp 200 TCSPC module (PicoQuant), and raw fluorescence lifetime images were collected by scanner at a 512 × 512 pixel resolution.

Images were analyzed using home-coded macros of Fiji Is Just ImageJ software. For the experiments performed on mitochondria, we used the MicroTime 200 fluorescence-lifetime microscope system (PicoQuant GmbH, Belin, Germany) described earlier with the same configuration, adding a bandpass filter (D520/40M, Chroma). The mitochondria biomarker used was the MitoTracker Green FM (ThermoFisher)

## 5. Conclusions

We studied the photophysical behavior of a new luminogen. We confirmed the AIE phenomenon with different approaches by using different solvent proportions, increasing the viscosity and forming micelles. In all cases, we observed a strong enhancement of the fluorescence produced by reducing the intrarotational mobility of the dye. Finally, we studied the rate of spontaneous incorporation of the dye into cells and observed the intracellular pattern produced by AIE. Interestingly, different intracellular compartments present strong differences in fluorescence intensity and fluorescence lifetime. We used this difference to isolate different intracellular regions to allow us to selectively study these regions. Finally, we have demonstrated the use of this luminogen in bioimaging with a mitochondria biomarker to analyze these organelles through their fluorescence lifetimes.

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
