# Peer review of "Synthesis, Photophysics, and Solvatochromic Studies of an Aggregated-Induced-Emission Luminogen Useful in Bioimaging"

_sensors, 2019, doi:10.3390/s19224932_

Round 1
Reviewer 1 Report
The work presents on novel phenomenon by well designed experiments. The manuscript is sometimes hard to read and contains repetitive statements on number of places. I suggest that authors discuss or compare their results on variations of cell autofluorescence lifetime, as it might be necessary to correct for such variations (figures 6 and 7) to make justifiable conclusions.
I recommend for publications after suggested minor corrections are included in the manuscript.
Reviewer 2 Report
Attached.

Reviewer 3 Report
The paper "Synthesis, photophysics and solvatochromic studies of an aggregated-induced-emission luminogen useful in bioimaging" aims to characterize the solvatofluorochromic properties of the dye PEMC, in response to different environments, to be then used to detect isolated intracellular compartments by imaging, depending on the response to possible different components.
In general, experiments need some revision, as well as the text, because of convoluted and repetitive sentences. Details are given below.
Therefore the paper needs a major revision, before to be considered suitable for publication in “Sensors”
Major remarks:
- 1) Line 36 “weak emission in dilute solutions and low quantum yield..”: one of the two is a repetition
- 2) Ref 1, on “…aggregation enhances the quantum yield and causes strong emission, we call this behavior aggregation-induced emission (AIE) …” Already in this point, it cannot be neglected that aggregation effects on dye fluorescence properties (in terms of wavelength- colour- and quantum yield) are reported since a long time on historical dyes (i.e. acridine orange; porphyrins). Also the sentence should recall changes in spectral position (i.e. shown for PEMC in Figure 1, and Table 1), besides in quantum yield.
- 3) Changes of spectral properties of PEMC depending on different solvents are shown in Figure 1, and data summarized in Table 1.
Given the importance of solvent polarity, also this information should be reported for each solvent .
Table 1 shows data from Nile red A, used as a reference standard. The calculation procedure of quantum yield should be described in detail.
- 4) the amount of PEMC administered to cells is correctly given as 5x10-6 M at line 172. Similarly, molar concentrations should be given at line 262.
- 5) line 174, the sentence “…fluorescence lifetime and this parameter is independent of the dye concentration..” can be misunderstanding. The concept needs to be clarified, because while “fluorescence lifetime .. is independent of the dye concentration” a sufficient level of intensity is obviously necessary to collect signals. In fact , Fig. 5 is “used” to show increase of intensity vs time.
Figure 6 reports on the detection/separation of different lifetimes able to isolate different cell organelles.
Identification of organelles matching with the three regions separated according to the lifetime should be appreciated, i.e. by means of colocalization with specific biomarkers, considering that the peripheral F-actin structures at the plasma membrane in HEK293 cell are indicated only on the basis of a reference.
- 6) Lines 187-190. Again concepts and results about intensity and lifetime are unclear. “…The difference in intensity can be used to isolate different organelles based on the intensity threshold (see Figure S6). This approach can efficiently study independent organelles based on fluorescence lifetime. Figure 7a shows HEK cells segmented by intensity and with the resulting fluorescence lifetime images maps and histograms...” Is seems that images are first sectioned depending on intensity level, and then for each level lifetimes are given? Please provide clearer sentences.
Round 2
Reviewer 2 Report
The authors have properly replied to all my queries and have made the corresponding changes to the manuscript.
Reviewer 3 Report
The paper "Synthesis, photophysics and solvatochromic studies of an aggregated-induced-emission luminogen useful in bioimaging" has been duly revised according to all the remarks from the Reviewer. Only some minor points still need attention as detailed below, before the paper can be considered suitable for publication in “Sensors”.
Minor remarks:
Table 1: Nile Blue A (ref), please specify the reference, revising the reference order if the case.
Identification of organelles: The answer on peripheral F-actin identified in terms of typical structures around the cells, as already reported in the literature and avoiding any mistake with other organelles is convincing, and for clearness the explanation should be added to the text.
Legend of Figure S3. ….effect and bottom without 1x10-7..”: Please check the meaning of “..bottom: “
Legend of Figure S9. “.. Fluorescence lifetime images maps of HEK cells with MitoTracker Green and PEMC (left) only PEMC (in the middle) and colocation region (Right)”.
“..only PEMC.. “ can be misunderstanding, suggesting a treatment with only PEMC, while this is due to the separation of red signal, is it? Please specify to make Legend clearer.
